# Center backs work hardest when playing in a back three: The influence of tactical formation on physical and technical match performance in professional soccer

**Leon Forcher** [1,2]*, **Leander Forcher**[1], **Darko Jekauc**[1], **Alexander Woll**[1], **Timo Gross**[2], **Stefan Altmann**[1,3]

1 Institute of Sports and Sports Science, Karlsruhe Institute of Technology, Karlsruhe, Germany, 2 TSG 1899 Hoffenheim, Zuzenhausen, Germany, 3 TSG ResearchLab gGmbH, Zuzenhausen, Germany

* leon.forcher@kit.edu

## Abstract

The purpose of this study was to investigate whether tactical formation affects the physical and technical match performance of professional soccer players in the first German Bundesliga. From official match data of the Bundesliga season 2018/19, physical (total distance, high-intensity distance, sprinting distance, accelerations, maximum velocity) and technical performance (short/middle/long passes, dribblings, ball-possessions) of players were analyzed. Players were categorized into five playing positions (center back, full back, central midfielder, wide midfielder, forward) and teams into eight different tactical formations (4-4-2, 4-4-2 diamond, 4-2-2-2, 4-3-3, 4-5-1, 4-2-3-1, 3-4-3, 3-5-2). Results revealed that the degree to which tactical formation affects match performance is position dependent. In terms of physical performance, center backs and full backs showed highest sprinting distances when playing in a formation with only three defenders in the back row (3-4-3, 3-5-2) compared to all other formations (ES range: $0.13 \leq ES \leq 1.27$). Regarding technical performance, all positions except forwards displayed fewer short passes, middle passes and ball-possessions in the formations 4-3-3 and 4-2-3-1 compared to all other formations ($0.02 \leq ES \leq 1.19$). In conclusion, physical and technical performance of center backs, full backs and wide midfielders differed markedly between the tactical formations. Conversely, the physical and technical performance of central midfielders and forwards only showed small differences between the different tactical formations. These findings can help coaches scheduling their practice. For example, if a coach wants to change the playing formation, he can anticipate the physical and technical match performance changes depending on the respective playing position.

## Introduction

The intensity and the speed of professional soccer have increased in recent years [1]. In favor of this development, the physical match performance of a player in a single match has risen

**Data Availability Statement:** The data that support the findings of this study are available from the Deutsche Fußball Liga (DFL). Restrictions apply to

the availability of these data, which were used under license for this study. The data is provided by a commercial company (Deltatre) and therefore the data is not freely available. Requests to access the datasets should be directed to the DFL (info@dfl.de). The authors received no special privileges in accessing the data from the DFL that other researchers would not have.

**Funding:** The funder TSG 1899 Hoffenheim provided support in the form of salaries for authors [LF, TG]. The funder TSG ResearchLab gGmbH provided support in the form of salaries for authors [SA]. Both funders did not have any additional role in the study design, data collection and analysis, decision to publish, or preparation of the manuscript. The specific roles of these authors are articulated in the 'author contributions' section.

**Competing interests:** The authors have read the journal's policy and have the following competing interests to declare: The authors [LF, TG] were employed by the commercial affiliation TSG 1899 Hoffenheim. The authors [SA] were employed by the non-commercial limited liability company TSG ResearchLab gGmbH. This does not alter our adherence to PLOS ONE policies on sharing data and materials. There are no patents, products in development or marketed products associated with this research to declare.

significantly [2]. Further, the technical skills that are required to compete on a professional level, have increased similarly [1, 3].

Looking at the performance of a soccer player, besides physical and technical parts, performance is also determined by mental and especially tactical aspects [4]. Among the most important tactical factors rank the playing position and the tactical formation.

The playing position has a large impact on the physical and technical match performance of a player [5, 6]. From a physical perspective, central midfielders show the highest total running distance compared to other positions [5–10]. Looking at the distances covered at high-intensity speed and sprinting speed, wide midfielders and full backs display greater distances than the other positions [5, 6, 9–13]. Regarding technical performance, Dellal et al. (2010) [5] revealed that forwards lose more duels and have more turnovers than other positional groups. Further, midfielders (central & wide) displayed the most ball-possessions.

The effect of tactical formation on match performance seems to be lower than the effect of playing position, however differences between formations have been revealed [14, 15]. One investigation showed higher amounts of passes played and success rate of passes for teams in a 4-4-2 formation compared with teams in a 4-3-3 or 4-5-1 formation [16]. Baptista et al. (2019) [14] revealed that players playing in a 4-5-1 formation covered more distance in high-intensity and sprinting speeds than in a 3-5-2 formation.

A drawback of the abovementioned studies is that they investigated the effects of tactical formation and playing position on match performance in isolation. Conversely, the combination of tatical formation and playing position seems more promising to explain match performance [7, 17].

Hence, some investigations tried to investigate the effects of the combination of the factors tactical formation and playing position on soccer match performance. A study that distinguished between the three positional groups defenders, midfielders, and attackers found that defenders showed lower total distance and high-intensity distance when playing in a 4-4-2 formation, compared to defenders in a 4-3-3 or 4-5-1 formation [16]. In addition, strikers cover a larger high-intensity distance when playing in a 4-3-3 formation, compared to strikers in a 4-4-2 or 4-5-1 formation. Building on these results, Tierney et al. [12] differentiated between five playing positions. Their findings revealed that central midfielders accelerate more often in the 4-2-3-1 formation and cover higher total and high-intensity distances in the 4-4-2 formation than central midfielders in other formations. Differentiating between center backs and wide defenders as well as between central and wide midfielders offered novel insights regarding the effect of tactical formation on soccer match performance. Only one investigation studied the combined effects of formation and position on the technical performance of soccer players [15], thereby analyzing how the tactical formation of the opposing team affects the technical match performance of one single soccer team. For example, it was found that central midfielders and center backs played more direct passes when playing against a team in a 4-2-3-1 formation, compared to opponents playing in a 4-4-2 formation.

While providing first insights into the combined effects of tactical formation and playing position on soccer match performance, the current state of research lacks findings of the influence on technical match performance. Furthermore, only a limited number of tactical formations (maximum 5 formations) have been investigated so far. Therefore, studies that capture all tactical formations used by teams from a whole league could provide a more comprehensive picture on this topic. Moreover, it is well known that the level and the origin of the league can impact the physical and technical match performance of soccer players [18, 19]. While there is no investigation addressing the German Bundesliga so far, it seems worthwhile to explore this topic in this league.

Therefore, the current study aimed to investigate whether tactical formation affects the physical and technical performance of professional soccer players of different positions in the

first German Bundesliga. Taking the results of other investigations into account [2, 20, 21], we hypothesized that according to the playing position, the formation affects the physical and technical performance.

## Materials and methods

### Sample

In the present study, official match data from the 2018/2019 season of the German Bundesliga were used, since this was the last season that has not been affected by the COVID-19 pandemic. A total of 267 out of 306 games were analyzed, as every match with one player has been sent off was excluded. Since only players that were involved in the whole game time (i.e., full 90 min) of the respective match were included, leading to a maximum of 20 outfield players per match. This results in 3810 separate observations (i.e. a single match performance of one player) that were analyzed. Although data was collected as part of the players' professional employment [22], ethical approval was obtained from the local ethics committee (Human and Business Sciences Institute, Saarland University, Germany, identification number: 22–02, 10 January 2022).

### Variables and procedures

Initially, the tactical formation for each team and match, respectively, was identified by using the official match-reports of the Bundesliga which are provided by Deltatre (Deltatre, Turin, Italy). The identified formations are constructed out of the starting eleven and are checked by observation after 15 minutes of each game. To investigate accuracy of the provided tactical formation data, we compared the formations provided for the first game day (18 formations) with the observation of an experienced video analyst of the German Bundesliga team TSG Hoffenheim. Given the high agreement between the results of the provided formations from Deltatre and the observations from the video analyst (Cohen's Kappa: 0.93, $p < 0.05$), the data from Deltatre were used for this study [23].

Additionally, five different playing positions were distinguished (central defender, full back, central midfielder, wide midfielder, forward). Subsequently, 9 different tactical formations differentiated (see S4 Table). As the formation 3-4-3 diamond was only played once, it was excluded from further analysis.

After identifying the tactical parameters formation and position, the physical and technical performance of the respective players were analyzed. To assess the physical performance, the parameters total distance [km], high-intensity distance [km], sprinting distance [km], the maximum velocity [km/h], and the number of accelerations [quantity] were analyzed. Considering the underlying data and the used speed zones of other studies [5, 8, 16, 19, 24], the speed interval for high-intensity distance was set for 17.00–23.99 km/h and sprinting distance set for ≥24.00 km/h. One acceleration was counted, when there was a positive acceleration score for more than 1,5 sec., implying there had to be an increase of speed compared to the frame before.

Technical performance was analyzed using the parameters number of passes, dribblings, and ball-possession phases. Based on the covered distance of the ball, passes were divided into three categories (short [<10 m], middle [10≥30 m], long [>30 m]). One dribbling was counted when one player in safe ball control tried to dribble past an opponent. One ball-possession phase for one player was counted when he had a ball action in a ball-possession phase of his team.

Finally, contextual factors that have been reported in other studies were analyzed for each match: Quality of the own team (= team ranking at the end of the season), quality of the

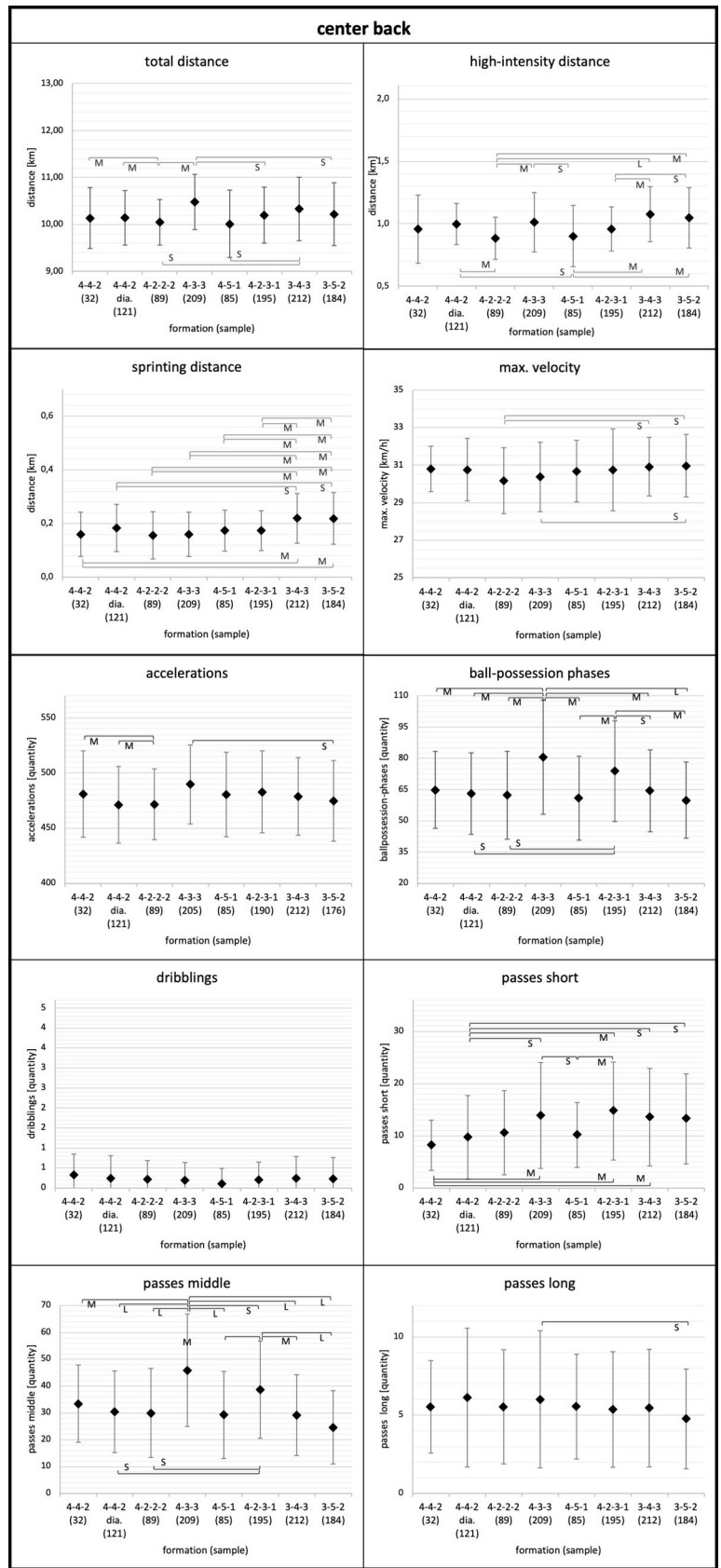

**Fig 1. Center back.** Data of center backs are presented as mean values ± SD. Anova revealed p<0.05 for each parameter except dribblings (p = 0.43). Black parentheses indicate significant differences (p<0.05) between the formations. Each significant group difference is labelled with S for small, M for medium or L for large effect size.

opponent (= team ranking at the end of the season) [9, 11], result of the game (= points in the respective game) [11], percentage of ball-possession [16], venue (home or away) [9, 11], and net playing time [25] were analyzed. These contextual factors were captured as they could possibly explain how tactical, physical, and technical factors interact with each other.

All data are based on the DFL Observed Tracking-Data, which are processed by Deltatre. The data were captured using a Multi-Camera-Tracking System (TRACAB, Chyron Hego, Melville, NY, USA), which can be considered valid [26].

## Statistical analysis

To analyze the effect of tactical formation within one playing position, each single playing position was considered independently. Therefore, for every single playing position (center back, full back, central midfielder, wide midfielder, forward) a one-way analysis of variance [ANOVA] was conducted separately for every physical (total distance, high-intensity distance, sprinting distance, max. velocity, accelerations) and technical (ballpossession-phases, dribblings, short/medium/long passes) parameter. In this context, the tactical formation served as the independent variable and the respective physical or technical parameter as the dependent variable. To determine possible differences between tactical formations, Bonferroni post-hoc tests were executed.

Further, the contextual factors were addressed individually. To check if the contextual factors differ according to the tactical formation, for each contextual factor (own team ranking, opposition team ranking, net game time, points per game, ball possession, venue) a one-way ANOVA was conducted. Similarly, the tactical formation served as the independent variable and the respective contextual parameter as the dependent variable. To determine possible differences between tactical formations, Bonferroni post-hoc tests were executed.

To interpret the magnitude of differences, Cohen's d effect sizes [ES] were computed: Small ($0.2 \leq ES < 0.5$), medium ($0.5 \leq ES < 0.8$) and large ($ES \geq 0.8$) ES were distinguished [27].

A priori, the significance for all tests was set to 0.05. All statistical analyses were executed using IBM SPSS Statistics 25.0.0.0 (IBM Co., New York, USA).

## Results

Means, standard deviations, and results for the ANOVA of the physical and technical parameters for each playing position considering the tactical formation are displayed in Figs 1–5. Descriptive values for each parameter can also be found in S1–S3 Tables. Overall, ANOVA revealed significant differences between tactical formations for all positions and regarding most physical and technical parameters (Figs 1–5).

More in detail, the degree to which tactical formation affected physical and technical match performance was position dependent. Relating to physical performance, center backs and full backs demonstrated the largest means for total and high-intensity distance in the 3-4-3 and 3-5-2 formations (Figs 1 and 2). Wide midfielders showed the highest values for total and high-intensity distance in the 4-4-2 diamond formation and the lowest values in the 3-4-3 formation (Fig 4). In addition, central midfielders and forwards displayed less pronounced differences in physical parameters (e.g. high-intensity distance) (Figs 3 and 5).

Concerning technical performance, full backs showed the highest amount in dribblings in 3-4-3 and 3-5-2 formations (Fig 2). By contrast, the number of dribblings for center backs and

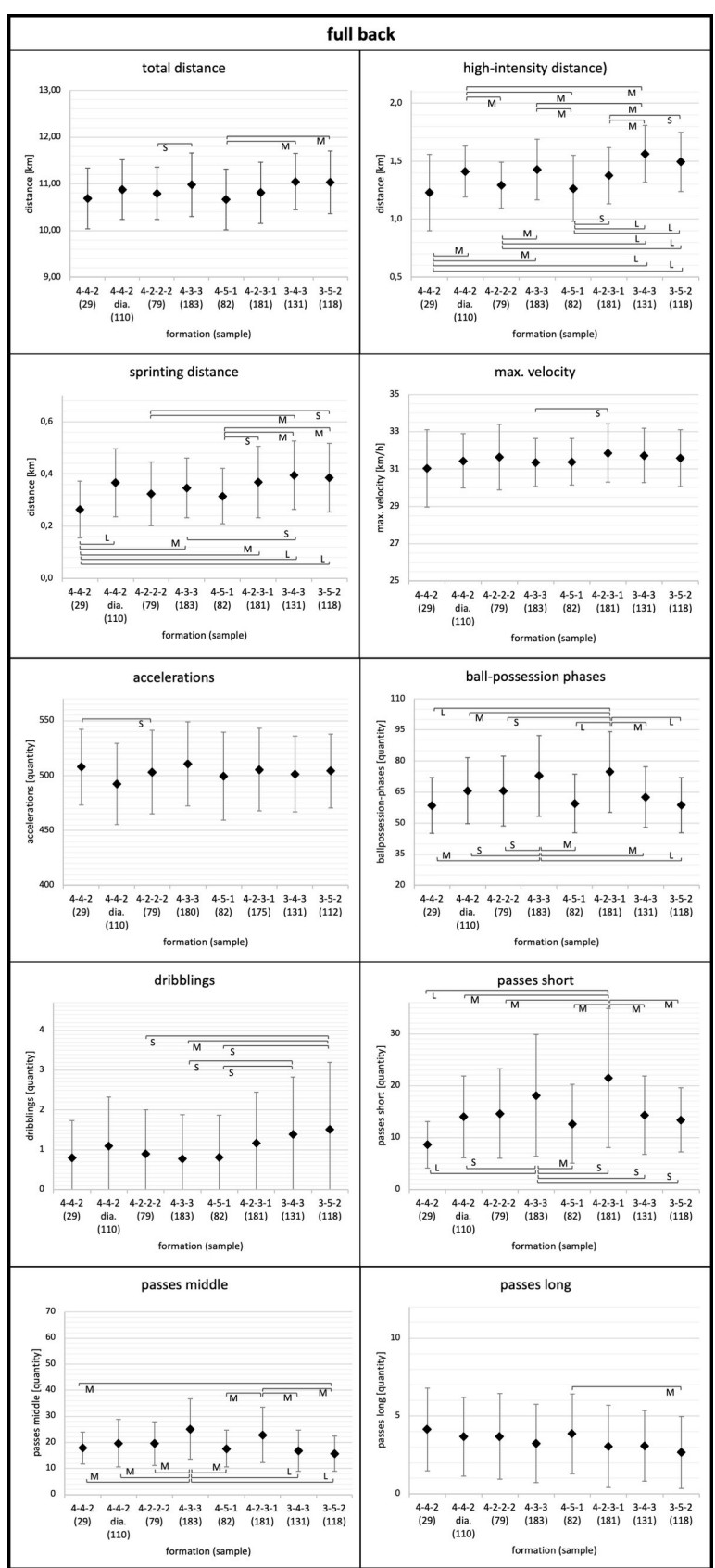

**Fig 2. Full back.** Data of full backs are presented as mean values ± SD. Anova revealed p<0.05 for each parameter. Black parentheses indicate significant differences (p<0.05) between the formations. Each significant group difference is labelled with S for small, M for medium or L for large effect size.

central midfielders were similar across formations (Figs 1 and 3). Except forwards, all other positions demonstrated higher values for short passes and ball-possession phases in the formations 4-3-3 and 4-2-3-1 (Figs 1–4).

Looking at the contextual factors, some of these parameters showed differences according to the tactical formation (Table 1). While opposition team ranking and venue were unaffected by the tactical formation, own team ranking and ball-possession differed markedly according to the tactical formation.

## Discussion

The study aimed to investigate whether tactical formation affects the physical and technical performance of professional soccer players of different positions in the first German Bundesliga.

The main finding was that the degree to which tactical formation affects match performance is position dependent. In this context, on the one hand, technical and physical performance of center backs, full backs and wide midfielders differed markedly between the tactical formations. On the other hand, the physical and technical performance of central midfielders and forwards only showed small differences between the different tactical formations. Therefore, the hypothesis that the tactical formation affects the physical and technical performance according to the playing position can be generally confirmed.

In the following, the results for each playing position will be discussed individually. Center backs demonstrated higher values for total distance and accelerations for the 4-3-3 formation compared to other formations (ES range: $0.19 \leq$ effect size [ES] $\leq 0.78$). This finding contradicts other investigations, which identified lower total distance and accelerations for center backs in 4-3-3 compared to other formations [12, 16, 28]. However, it should be noted, that these investigations used relatively small sample sizes which might limit their explanatory power. Further, considering the high-intensity distance, center backs showed the highest values in 4-3-3, 3-4-3, and 3-5-2. Compared to other formations, there was a range from small to large differences ($0.06 \leq ES \leq 0.93$). Similarly, center backs covered more sprinting distance in 3-4-3 and 3-5-2 compared to all other formations ($0.38 \leq ES \leq 0.70$). Other researchers also found higher sprinting distances for center backs in a 3-5-2 formation [14, 24]. The results could be associated with the assumption that in 3-4-3 and 3-5-2 formations, full backs can be more offensive as three center backs ensure higher defensive protection compared to formations with only two center backs. Therefore, only three center backs have to cover the length and the width of the field, while in other formations (e.g. 4-4-2) there are four players to do so.

Concerning the technical performance, center backs showed higher values for ball-possession phases, short passes, and middle passes for 4-3-3 und 4-2-3-1 compared to other tactical formations ($0.03 \leq ES \leq 1.19$). A possible explanation for the increased ball-possession phases of center backs might be that in the 4-3-3 and 4-2-3-1 formations, the contextual factor ball-possession per team was higher than in other formations (see Table 1). Moreover, a higher percentage of ball-possession enables the respective players (e.g. center backs) to complete more passes.

Full backs, in general, showed a more straightforward response in physical performance between tactical formations. On the one hand, lowest total distance, high-intensity distance, and sprinting distance were observed in the formations 4-4-2 and 4-5-1. On the other hand,

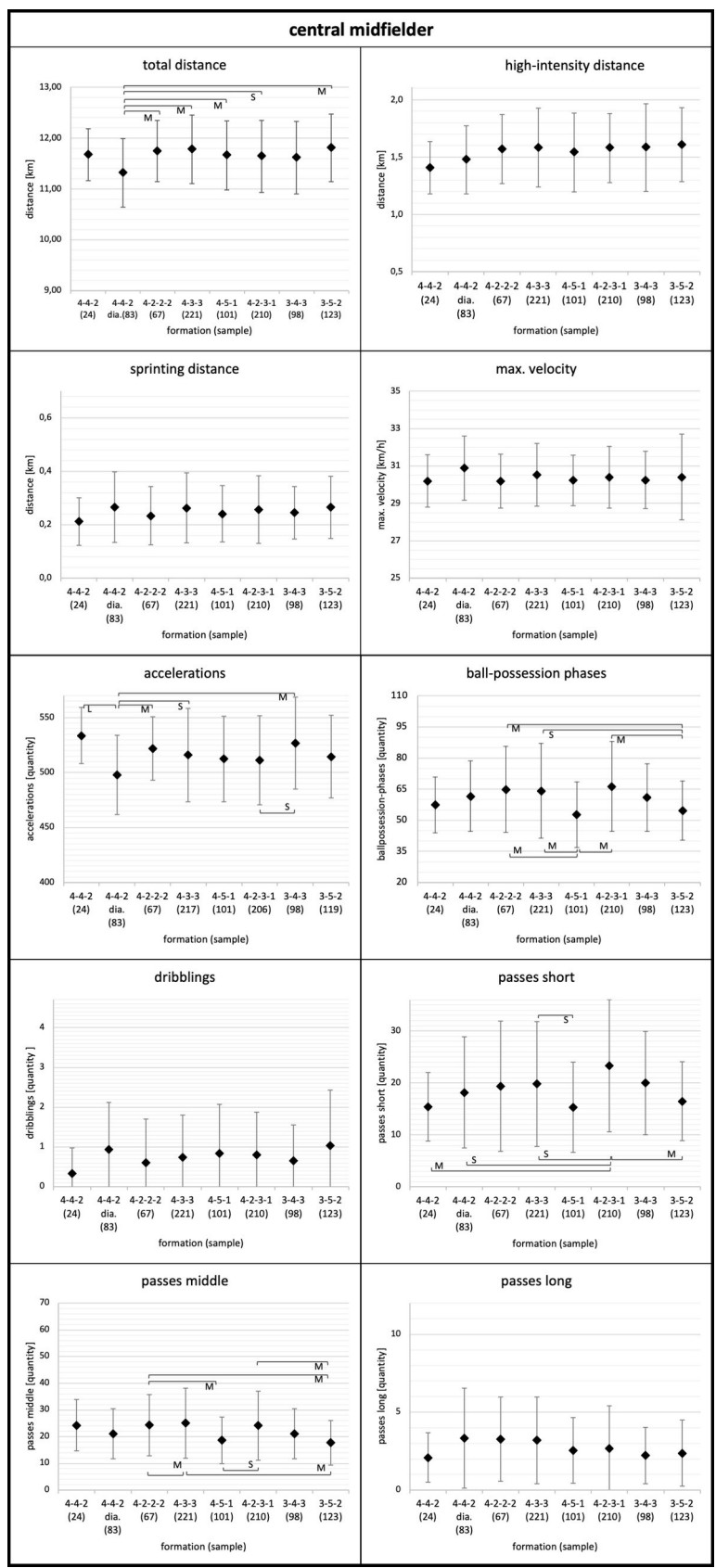

**Fig 3. Central midfielder.** Data of central midfielders are presented as mean values ± SD. Anova revealed p<0.05 for each parameter except sprinting distance (p = 0.20) and maximum velocity (p = 0.14). Black parentheses indicate significant differences (p<0.05) between the formations. Each significant group difference is labelled with S for small, M for medium or L for large effect size.

greatest total, high-intensity, and sprinting distances were found for 3-4-3 and 3-5-2 with up to large effect sizes in comparison with other formations (0.13≤ES≤1.27). Supporting these results, a study by Modric et al. [24] revealed highest values for total, high-intensity, and sprinting distances for full backs in a 3-5-2 formation. Therefore, based on our and Modric and colleague's [24] findings, full backs show a higher running performance (i.e. total distance, high-intensity distance, sprinting distance) in formations with three center backs compared to formations with four defenders (e.g. 4-4-2 or 4-5-1). An explanatory approach could be that full backs receive more defensive support in 3-4-3 and 3-5-2 formations by the three center backs and therefore can focus more on their offensive duties. This results in more running output for the full backs to fulfill their offensive and defensive responsibilities.

Looking at the technical performance, full backs displayed more dribblings in 3-4-3 and 3-5-2 compared with other formations (0.16≤ES≤0.54). This could be related to the explanatory approach that full backs have more offensive responsibilities in formations with three center backs. Full backs in 3-4-3 and 3-5-2 act in more offensive positions and therefore can attempt more dribblings. Full backs also show higher values for ball-possession phases, short passes, and middle passes in 4-3-3 and 4-2-3-1 compared to other formations (0.31≤ES≤1.02). As mentioned earlier, these results can be related to the contextual factor of ball-possession. Further, the teams playing 4-3-3 and 4-2-3-1 had a higher team ranking compared to other formations (see Table 1). In this context, an investigation revealed that better teams more often played a ball-possession-based style [29]. These findings indicate that the results of ball-possession percentage and quality of a team can be related to each other.

Considering the physical performance of central midfielders, only a few differences occur between formations. Central midfielders in 4-4-2 diamond exhibit a lower running performance (i.e. total distance, high-intensity distance, sprinting distance) compared to other formations. Other investigations revealed more pronounced differences for central midfielders between formations. However, these studies only looked at data of one or two teams with relatively small sample sizes, therefore restricting their findings [12, 14].

Similarly, there only occurred a few differences between formations in technical parameters. As mentioned above, central midfielders are more involved in ball possessions in 4-3-3 and 4-2-3-1 formations. Therefore, they exhibited more short and middle distance passes in these formations. Again, this could be related to the contextual factors of team ranking and ball-possession. Due to the central positioning in all formations, central midfielders potentially do not have to adapt their physical and technical performance as much as other positions (center back, full back) when changing the tactical formation.

Regarding the position wide midfielder, more differences than for central midfielders were discovered. Higher values were found for wide midfielders in 4-4-2 diamond formation in the total and high-intensity distance and lower values for sprinting distance compared to other formations (0.16≤ES≤1.36). Furthermore, wide midfielder in a 3-4-3 formation experienced a smaller physical load than wide midfielder in other formations. More specifically, wide midfielders showed lower values in 3-4-3 formation for total distance, high-intensity distance, sprinting distance, and accelerations compared to other formations (0.13≤ES≤1.36). By contrast, other investigations were not able to reveal a smaller load for wide midfielders in a 3-4-3 formation [12]. However, Tierney et al. used data from two youth teams, and therefore the results are not comparable to those of the present study.

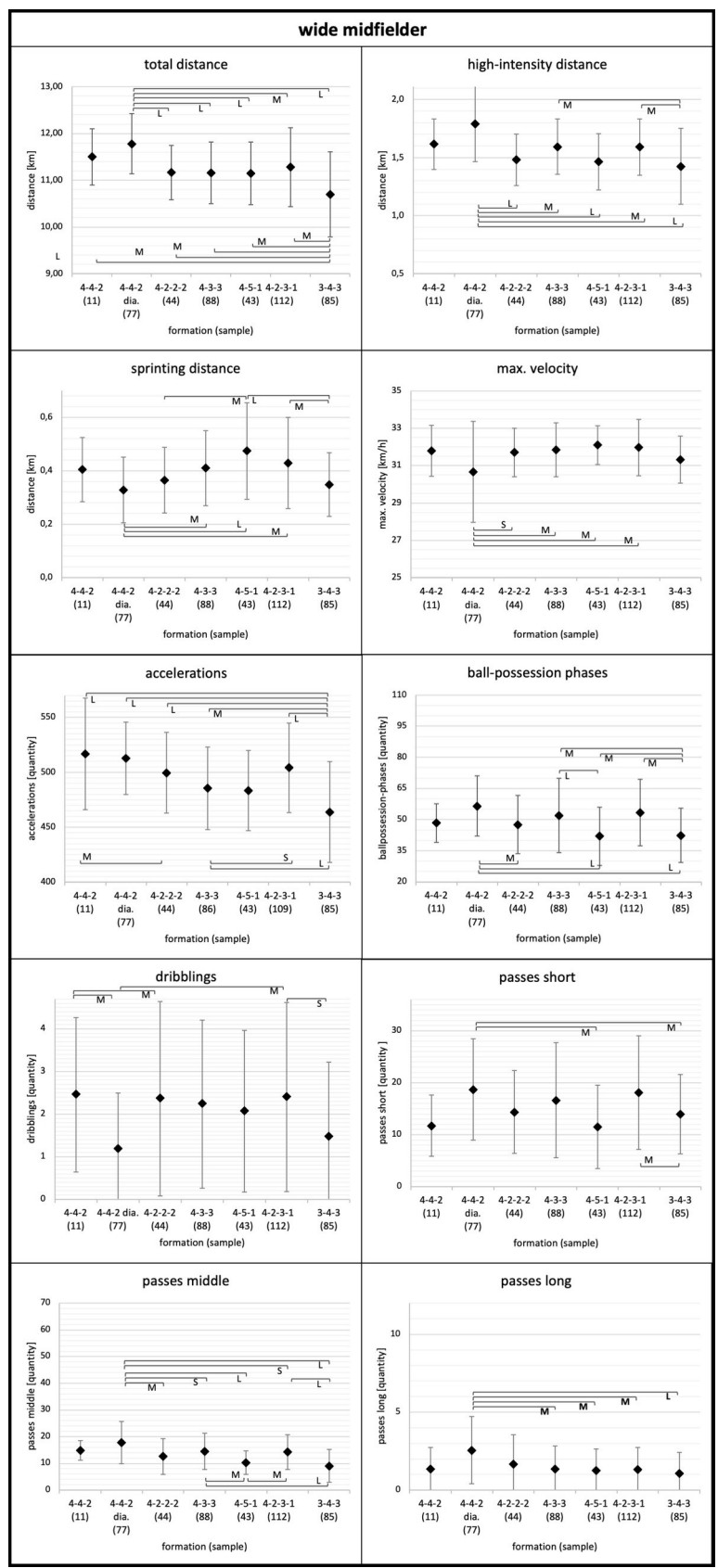

**Fig 4. Wide midfielder.** Data of wide midfielders are presented as mean values ± SD. Anova revealed p<0.05 for each parameter. Black parentheses indicate significant differences (p<0.05) between the formations. Each significant group difference is labelled with S for small, M for medium or L for large effect size.

Additionally, wide midfielders showed more ball possessions, short, middle, and long passes as well as fewer dribblings in the 4-4-2 diamond formation compared to other formations (0.06≤ES≤1.25). The technical as well as the physical performance of wide midfielders in 4-4-2 diamond are similar to the general match-performance profile of central midfielders (see S3 Table). Therefore, it is reasonable to conclude that wide midfielders act similar to central midfielders due to their central positioning in the diamond formation. Similarly, higher values for ball possessions, short and middle passes were evident in the formations 4-2-3-1 and 4-3-3. As mentioned previously, this finding could be related to different contextual factors (ball possession, team ranking).

Regarding forwards, there were only little differences between the formations in terms of physical performance. Contrasting the results of several other investigations [12, 14, 24, 28] that found the highest total distance for forwards in 3-5-2, the present results revealed the lowest total distance for forwards in 3-5-2. Furthermore, forwards in the 4-4-2 diamond formation showed higher values regarding sprint distance and maximum speed compared to other formations (0.40≤ES≤1.09). These two parameters (sprinting distance, maximum speed) could probably be associated with each other. Larger sprinting distances of a player are associated with either longer distances per sprint or a higher number of sprints. In both cases, the chance of a higher maximum speed potentially increases.

Regarding the technical performance, there is no clear tendency identifiable. It is worth noting that forward is the only position where no higher values were found for middle passes and ball-possessions in 4-2-3-1 und 4-3-3. The position of forwards is higher up on the pitch compared to the other positions. Thus, they do not benefit from higher ball-possession percentages of their team, which commonly not manifest in the attacking third.

There are some limitations that need to be acknowledged, with the first relating to the sample of players. In detail, only players were included that participated in the whole specific match. Since offensive players are substituted more frequently, this results in a smaller sample size for these positions [30]. Furthermore, only starters are included and the results are not transferable to substitutes. Moreover, the Bundesliga increased the possible amount of substitutions from three up to five in relation to the COVID-19 pandemic. Therefore, one could assume that the impact of substitutions has increased because of the rule change. This topic needs to be addressed in future studies. In addition, the tactical formations and the playing positions were recorded at the beginning (first 15 minutes) of each game. Therefore, possible position and formation changes could not be considered. The positions and playing formations indeed were reviewed by a game analyst of a German Bundesliga team but still can only represent a reduced picture of reality. Another limitation regarding the statistical analysis with ANOVAs is present. In the present study, game observations of some players could potentially be included in different groups and hence the groups cannot be considered completely independent. Therefore, the analysis with ANOVAs might not be optimal. However, other approaches such as mixed models do not provide analysis of group differences considering the current research question. Therefore, despite the inherent limitations, ANOVAs were applied as they provide robust and conservative analysis of group differences. To help this problem, we provided effect sizes to help interpret the restricted results of the ANOVAs. Nevertheless, it is fundamental to further explore the combined effects of tactical formation and position on physical and technical match performance in soccer.

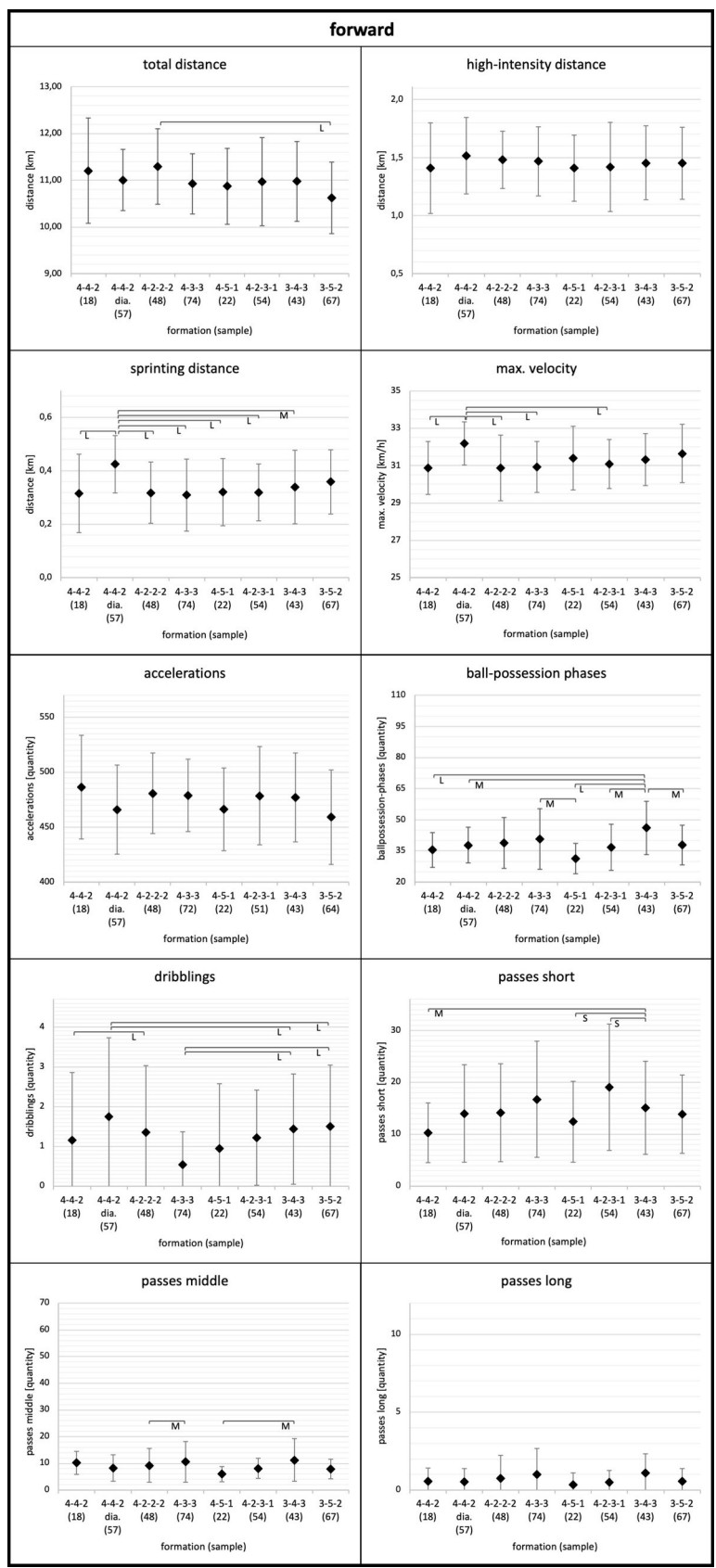

**Fig 5. Forward.** Data of forwards are presented as mean values ± SD. Anova revealed p<0.05 for each parameter except high-intensity distance (p = 0.80). Black parentheses indicate significant differences (p<0.05) between the formations. Each significant group difference is labelled with S for small, M for medium or L for large effect size.

Regarding future studies, investigating other leagues seems crucial given that match performance is dependent on the competitive level and the country [18, 19]. To allow for comparison between studies, standardized coding of positions and formations seems fruitful. In addition, most studies only looked at physical performance and therefore, technical aspects should get more attention in upcoming studies.

**Table 1.**

| Formation | games | mean | SD | anova | group comparisons |
|---|---|---|---|---|---|
| **own team ranking (end of the season)** | | | | | |
| 4-4-2 | 16 | 13.50 | 2.48 | p<0.01 | [***vs. 4-3-3]; [***vs. 4-2-3-1] |
| 4-4-2 dia. | 63 | 9.70 | 3.99 | | [**vs. 4-3-3]; [***vs. 4-5-1]; |
| 4-2-2-2 | 46 | 10.50 | 5.72 | | [***vs. 4-3-3]; |
| 4-3-3 | 109 | 6.38 | 4.50 | | [***vs. 4-4-2]; [**vs. 4-4-2 dia.]; [***vs. 4-2-2-2]; [**vs. 4-2-3-1]; [***vs. 4-5-1]; [***vs. 3-4-3]; [***vs. 3-5-2] |
| 4-5-1 | 46 | 13.43 | 4.01 | | [***vs. 4-4-2 dia.]; [***vs. 4-3-3]; [***vs. 4-2-3-1]; [**vs. 3-5-2] |
| 4-2-3-1 | 106 | 7.53 | 5.63 | | [***vs. 4-4-2]; [**vs. 4-2-2-2]; [***vs. 4-5-1]; [**vs. 3-4-3]; [**vs. 3-5-2] |
| 3-4-3 | 78 | 11.12 | 4.16 | | [***vs. 4-3-3]; [***vs. 4-2-3-1] |
| 3-5-2 | 69 | 10.55 | 4.37 | | [***vs. 4-3-3]; [**vs. 4-5-1]; [***vs. 4-2-3-1] |
| **opposition team ranking (end of the season)** | | | | | |
| 4-4-2 | 16 | 8.44 | 5.27 | p = 0.16 | no significant differences between formations |
| 4-4-2 dia. | 63 | 9.70 | 4.78 | | no significant differences between formations |
| 4-2-2-2 | 46 | 10.67 | 4.94 | | no significant differences between formations |
| 4-3-3 | 109 | 9.71 | 5.09 | | no significant differences between formations |
| 4-5-1 | 46 | 7.70 | 5.41 | | no significant differences between formations |
| 4-2-3-1 | 106 | 9.86 | 5.09 | | no significant differences between formations |
| 3-4-3 | 78 | 9.55 | 5.15 | | no significant differences between formations |
| 3-5-2 | 69 | 8.83 | 5.68 | | no significant differences between formations |
| **net game time [min]** | | | | | |
| 4-4-2 | 16 | 58.91 | 4.38 | p<0.01 | no significant differences between formations |
| 4-4-2 dia. | 63 | 56.23 | 3.94 | | [**vs. 4-3-3] |
| 4-2-2-2 | 46 | 56.98 | 4.19 | | no significant differences between formations |
| 4-3-3 | 109 | 58.73 | 4.25 | | [**vs. 4-4-2 dia.]; [**vs. 3-4-3]; [**vs. 3-5-2] |
| 4-5-1 | 46 | 57.84 | 3.90 | | no significant differences between formations |
| 4-2-3-1 | 106 | 58.30 | 4.65 | | no significant differences between formations |
| 3-4-3 | 78 | 56.46 | 4.00 | | [**vs. 4-3-3] |
| 3-5-2 | 69 | 56.32 | 3.91 | | [**vs. 4-3-3] |
| **points per game [quantity]** | | | | | |
| 4-4-2 | 16 | 1.00 | 1.26 | p<0.01 | no significant differences between formations |
| 4-4-2 dia. | 63 | 1.71 | 1.33 | | [**vs. 4-5-1]; [**vs. 3-4-3] |
| 4-2-2-2 | 46 | 1.67 | 1.38 | | no significant differences between formations |
| 4-3-3 | 109 | 1.51 | 1.33 | | no significant differences between formations |
| 4-5-1 | 46 | 0.87 | 1.20 | | [**vs. 4-4-2 dia.]; [**vs. 4-2-3-1] |
| 4-2-3-1 | 106 | 1.68 | 1.35 | | [**vs. 4-5-1]; [**vs. 3-4-3] |
| 3-4-3 | 78 | 0.97 | 1.23 | | [**vs. 4-4-2 dia.]; [**vs. 4-2-3-1] |
| 3-5-2 | 69 | 1.17 | 1.21 | | no significant differences between formations |

*(Continued)*

**Table 1.** (Continued)

| Formation | games | mean | SD | anova | group comparisons |
|---|---|---|---|---|---|
| **ball-possession [%]** | | | | | |
| 4-4-2 | 16 | 45.55 | 6.37 | p<0.01 | [***vs. 4-3-3] |
| 4-4-2 dia. | 63 | 50.05 | 7.35 | | [**vs. 4-5-1] |
| 4-2-2-2 | 46 | 48.09 | 8.17 | | [**vs. 4-3-3] |
| 4-3-3 | 109 | 53.92 | 9.13 | | [***vs. 4-4-2]; [**vs. 4-2-2-2]; [***vs. 4-5-1]; [***vs. 3-5-2] |
| 4-5-1 | 46 | 44.32 | 8.32 | | [**vs. 4-4-2 dia.]; [***vs. 4-3-3]; [***vs. 4-2-3-1]; [**vs. 3-4-3] |
| 4-2-3-1 | 106 | 51.98 | 8.99 | | [***vs. 4-5-1]; [**vs. 3-5-2] |
| 3-4-3 | 78 | 50.09 | 8.01 | | [**vs. 4-5-1] |
| 3-5-2 | 69 | 46.63 | 7.65 | | [***vs. 4-3-3]; [**vs. 4-2-3-1] |
| **venue (home [1] / away [2])** | | | | | |
| 4-4-2 | 16 | 1.50 | 0.52 | p>0.99 | no significant differences between formations |
| 4-4-2 dia. | 63 | 1.49 | 0.50 | | no significant differences between formations |
| 4-2-2-2 | 46 | 1.50 | 0.51 | | no significant differences between formations |
| 4-3-3 | 109 | 1.50 | 0.50 | | no significant differences between formations |
| 4-5-1 | 46 | 1.46 | 0.50 | | no significant differences between formations |
| 4-2-3-1 | 106 | 1.52 | 0.50 | | no significant differences between formations |
| 3-4-3 | 78 | 1.50 | 0.50 | | no significant differences between formations |
| 3-5-2 | 69 | 1.51 | 0.50 | | no significant differences between formations |

dia. = diamond

Data of contextual factors are presented as mean values ± SD. Significant group differences (p<0.05) are presented with small effect size *, medium effect size** and large effect size ***.

## Conclusion

This study revealed that tactical formation affects physical and technical match performance of professional soccer players. Moreover, the changes in match performance differ according to the specific playing position.

Physical and technical performance of center backs, full backs and wide midfielders differed markedly between the tactical formations. For example, center backs and full backs showed higher physical performance when playing in a formation with three defenders in the back row (3-4-3 & 3-5-2). Due to the central positioning in the 4-4-2 diamond formation, in this formation, wide midfielders showed physical and technical performance similar to the general profile of central midfielders. Conversely, central midfielders and forwards demonstrated less pronounced differences between different formations regarding the physical and technical match performance.

From a practical point of view, results can help coaches in scheduling their practice. For example, if a coach wants to change the playing formation he can anticipate the changes in physical and technical load for each playing position and can adapt training and recovery processes accordingly.

## Supporting information

**S1 Table. Descriptive values (mean ± SD) per position (center back, full back, central midfielder, wide midfielder, forward) depending on the tactical formation.**
(DOCX)

**S2 Table. Descriptive values (mean ± SD) depending on the tactical formation.**
(DOCX)

**S3 Table. Descriptive values (mean ± SD) depending on the playing position.**
(DOCX)

**S4 Table. Number of players per position (center back, full back, central midfielder, wide midfielder, forward) depending on the tactical formation.**
(DOCX)

**S1 File. Distribution of the playing positions in the different tactical formations.**
(PDF)

## Acknowledgments

The authors thank the Deutsche Fußball Liga (DFL) for providing the match data used in this study. Further, there are no patents, products in development or marketed products associated with this research to declare. The data that support the findings of this study are available from the Deutsche Fußball Liga (DFL). Restrictions apply to the availability of these data, which were used under license for this study. The data is provided by a commercial company (Deltatre) and therefore the data is not freely available. Requests to access the datasets should be directed to the DFL (info@dfl.de).

## Author Contributions

**Conceptualization:** Leon Forcher, Leander Forcher, Darko Jekauc, Alexander Woll, Timo Gross, Stefan Altmann.

**Data curation:** Leon Forcher, Darko Jekauc.

**Formal analysis:** Leon Forcher.

**Investigation:** Leon Forcher, Leander Forcher.

**Methodology:** Leon Forcher, Leander Forcher, Darko Jekauc, Alexander Woll, Timo Gross, Stefan Altmann.

**Resources:** Timo Gross, Stefan Altmann.

**Supervision:** Darko Jekauc, Alexander Woll, Stefan Altmann.

**Validation:** Leon Forcher, Stefan Altmann.

**Visualization:** Leon Forcher.

**Writing – original draft:** Leon Forcher.

**Writing – review & editing:** Leon Forcher, Leander Forcher, Darko Jekauc, Alexander Woll, Timo Gross, Stefan Altmann.

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
