## [Decision Letter · Decision Letter 0]

16 Dec 2021

PONE-D-21-27534Center Backs Work Hardest When Playing in a Back Three: The Influence of Tactical Formation on Physical and Technical Match Performance in Professional SoccerPLOS ONE

Dear Dr. Forcher,

Thank you for submitting your manuscript to PLOS ONE. After careful consideration, we feel that it has merit but does not fully meet PLOS ONE’s publication criteria as it currently stands. Therefore, we invite you to submit a revised version of the manuscript that addresses the points raised during the review process.

Your manuscript was reviewed by two experts whose comments appear below. You will see that both reviewers have a positive impression of your work. However, at the same time, they have raised (particularly #2) important issues related to your methodology, manuscript organization, and statistical analyses that prevent publication in the present form. We will need you to address all of the points raised to proceed with the publication further.

We look forward to receiving your revised manuscript.

Kind regards,

Haroldo V. Ribeiro

Academic Editor

PLOS ONE

“The funder TSG 1899 Hoffenheim provided support in the form of salaries for authors [LF, TG]. The funder TSG ResearchLab gGmbH provided support in the form of salaries for authors [SA]. Both funders did not have any additional role in the study design, data collection and analysis, decision to publish, or preparation of the manuscript. The specific roles of these authors are articulated in the ‘author contributions’ section.”

“The funder TSG 1899 Hoffenheim provided support in the form of salaries for authors [LF, TG]. The funder TSG ResearchLab gGmbH provided support in the form of salaries for authors [SA]. Both funders did not have any additional role in the study design, data collection and analysis, decision to publish, or preparation of the manuscript. The specific roles of these authors are articulated in the ‘author contributions’ section.”

“The authors have read the journal's policy and have the following competing interests to declare: The authors [LF, TG] were employed by the commercial affiliation TSG 1899 Hoffenheim. The authors [SA] were employed by the non-commercial limited liability company TSG ResearchLab gGmbH. This does not alter our adherence to PLOS ONE policies on sharing data and materials. There are no patents, products in development or marketed products associated with this research to declare.”

6. We note that you have indicated that data from this study are available upon request. PLOS only allows data to be available upon request if there are legal or ethical restrictions on sharing data publicly. For more information on unacceptable data access restrictions, please see http://journals.plos.org/plosone/s/data-availability#loc-unacceptable-data-access-restrictions.

7. Please note that in order to use the direct billing option the corresponding author must be affiliated with the chosen institute. Please either amend your manuscript to change the affiliation or corresponding author, or email us at plosone@plos.org with a request to remove this option.

8. Please include your full ethics statement in the ‘Methods’ section of your manuscript file. In your statement, please include the full name of the IRB or ethics committee who approved or waived your study, as well as whether or not you obtained informed written or verbal consent. If consent was waived for your study, please include this information in your statement as well.

Reviewers' comments:

Reviewer's Responses to Questions

**Comments to the Author**

1. Is the manuscript technically sound, and do the data support the conclusions?

Reviewer #1: Yes

Reviewer #2: No

2. Has the statistical analysis been performed appropriately and rigorously? 

Reviewer #1: Yes

Reviewer #2: No

3. Have the authors made all data underlying the findings in their manuscript fully available?

Reviewer #1: No

Reviewer #2: Yes

4. Is the manuscript presented in an intelligible fashion and written in standard English?

Reviewer #1: Yes

Reviewer #2: Yes

5. Review Comments to the Author

Reviewer #1: The authors studied how the tactical formation in soccer could affect different positions' technical and physical performance.

The manuscript is straightforward to read if the reader knows about soccer and statistical analysis. Maybe soccer's fans and specialized commentators are not always familiar with statistical analysis. Perhaps the curious scientific reader is not always familiar with soccer team formations like 4-4-2 or 5-3-2-1.

The recommendation is minor revisions:

Therefore, according to the previous paragraph, the suggestions are:

1) Make pictures to illustrate the different tactical formations, and identify the positions (for example, the "full back," "center back,"...). The graphical representation in soccer broadcasts and specialized websites is pretty good and could enhance the clarity.

2) The Statistical Analysis section could present more detailed information about the methods; in this case, keep in mind the soccer enthusiasts that may be not familiar with acronyms like "one-way ANOVA" or even a brief explanation about "mean" and "standard deviations."

3) The data are previous to the Covid-19 pandemic. A meaningful change that occurred was the adoption of 5 substitutions per match, and there is the possibility that it becomes permanent, mainly due to the health concerns of the players. And this is a good point to discuss, mainly because this work could be compared (in the future) with this new scenario of 5 substitutions.

Reviewer #2: Report of "Center Backs Work Hardest When Playing in a Back Three: The Influence of

3 Tactical Formation on Physical and Technical Match Performance in Professional

4 Soccer" by L. Forcher et al.

The manuscript above presents a study of whether tactical formation affects the physical and 42 technical match performance of professional soccer players in the first German Bundesliga. While the topic of the manuscript is interesting, there are certain aspects that need to be clarified further, specially the methodology and robustness of the findings, before this manuscript is suitable for publication. Please find my detailed points below.

1) My main concern is that the most important results were extracted using ANOVA software, which

provides limited information, specially to evaluate the robustness of the findings. It seems that the authors

used the ANOVA as a black-box without examining details in the calculations.

2) The manuscripts present several issues regarding the organization and final presentation. How do you cite? Did you put number or names in parentheses? Lines 70, 72 and 73 (#). Names, lines 155 and 156 (names). Capital letters: lines 191, 192 and 194, figure captions; figure, table,

3) In the introduction they do not say how the work is organized.

4) The authors must consider additional metrics to present their results. The information provided by Deltatre can be used to explore additional properties, for instance, cross-correlations, etc.

5) The authors should also consider to incorporate methodologies like complex networks to reinforce their analysis and not just use conventional statistics.

6) As a general remark, it seems that they only make a statistical analysis but not checked or confirmed with anything else. The explanation of methods is not clear at all and it is not well understood the procedure. Also the quality of figures is very poor.

6. PLOS authors have the option to publish the peer review history of their article (what does this mean?). If published, this will include your full peer review and any attached files.

Reviewer #1: No

Reviewer #2: No

---

## [Author Response · Author response to Decision Letter 0]

28 Jan 2022

PONE-D-21-27534, entitled 

"Center Backs Work Hardest When Playing in a Back Three: The Influence of Tactical Formation on Physical and Technical Match Performance in Professional Soccer"

Submitted to: PLOS ONE

Decision: Major Revisions

Point-by-point response

Dear Editor,

Dear Reviewers,

The authors would like to thank the editor and the reviewers for their careful consideration and constructive criticism of this manuscript. We appreciate their positive comments, at the same time, agree with their suggestions for change, and have now revised our manuscript accordingly. Please find below a detailed point-by-point response.

The Authors

 

JOURNAL REQUIREMENTS:

Thank you for this comment. We updated every aspect of the manuscript and file formatting that is required. (see page 1-19)

We did not receive any third-party funds. Further, we updated the ‘Funding Information’ in the Cover letter.

Funding Statement: “The funder TSG 1899 Hoffenheim provided support in the form of salaries for authors [LeoF, TG]. The funder TSG ResearchLab gGmbH provided support in the form of salaries for authors [SA]. Both funders did not have any additional role in the study design, data collection and analysis, decision to publish, or preparation of the manuscript. The specific roles of these authors are articulated in the ‘author contributions’ section.”

“The funder TSG 1899 Hoffenheim provided support in the form of salaries for authors [LF, TG]. The funder TSG ResearchLab gGmbH provided support in the form of salaries for authors [SA]. Both funders did not have any additional role in the study design, data collection and analysis, decision to publish, or preparation of the manuscript. The specific roles of these authors are articulated in the ‘author contributions’ section.”

“The funder TSG 1899 Hoffenheim provided support in the form of salaries for authors [LF, TG]. The funder TSG ResearchLab gGmbH provided support in the form of salaries for authors [SA]. Both funders did not have any additional role in the study design, data collection and analysis, decision to publish, or preparation of the manuscript. The specific roles of these authors are articulated in the ‘author contributions’ section.”

We added the updated version of the ‘Funding Statement’ to the cover letter and excluded any funding information from the manuscript.

“The authors have read the journal's policy and have the following competing interests to declare: The authors [LF, TG] were employed by the commercial affiliation TSG 1899 Hoffenheim. The authors [SA] were employed by the non-commercial limited liability company TSG ResearchLab gGmbH. This does not alter our adherence to PLOS ONE policies on sharing data and materials. There are no patents, products in development or marketed products associated with this research to declare.”

We excluded any information about competing interests from the manuscript and added the updated version of the ‘Funding Statement’ to the cover letter.

We have indicated that the data from this study are available upon request because of legal restrictions by the data provider (DFL – Deutsche Fußball Liga). Requests to access the datasets should be directed to the DFL (info@dfl.de). The minimal data set we can provide can be found in the Supporting Information (S1 Table). 

6. We note that you have indicated that data from this study are available upon request. PLOS only allows data to be available upon request if there are legal or ethical restrictions on sharing data publicly. For more information on unacceptable data access restrictions, please see http://journals.plos.org/plosone/s/data-availability#loc-unacceptable-data-access-restrictions.

We have indicated that the data from this study are available upon request because of legal restrictions by the data provider (DFL – Deutsche Fußball Liga). 

7. Please note that in order to use the direct billing option the corresponding author must be affiliated with the chosen institute. Please either amend your manuscript to change the affiliation or corresponding author, or email us at plosone@plos.org with a request to remove this option.

The corresponding author is affiliated with the chosen institute (see manuscript – title page).

8. Please include your full ethics statement in the ‘Methods’ section of your manuscript file. In your statement, please include the full name of the IRB or ethics committee who approved or waived your study, as well as whether or not you obtained informed written or verbal consent. If consent was waived for your study, please include this information in your statement as well.

We provided an ethics statement in the methods section. TO further help the understanding we changed this paragraph into the following: “Although data was collected as part of the players’ professional employment (22), ethical approval was obtained from the local ethics committee (Human and Business Sciences Institute, Saarland University, Germany, identification number: 22-02, 10 January 2022).” The cited article suggests that in the present setting (professional soccer players) an ethical approval is not required. (See page 5, lines 134-136)

Reviewer #1: The authors studied how the tactical formation in soccer could affect different positions' technical and physical performance.

The manuscript is straightforward to read if the reader knows about soccer and statistical analysis. Maybe soccer's fans and specialized commentators are not always familiar with statistical analysis. Perhaps the curious scientific reader is not always familiar with soccer team formations like 4-4-2 or 5-3-2-1.

The recommendation is minor revisions:

Therefore, according to the previous paragraph, the suggestions are:

1) Make pictures to illustrate the different tactical formations, and identify the positions (for example, the "full back," "center back,"...). The graphical representation in soccer broadcasts and specialized websites is pretty good and could enhance the clarity.

Thank you for the comment. We added images of the different tactical formations to the Supporting Information (See S5 File). In the added pictures the positions in the tactical formations become clear.

2) The Statistical Analysis section could present more detailed information about the methods; in this case, keep in mind the soccer enthusiasts that may be not familiar with acronyms like "one-way ANOVA" or even a brief explanation about "mean" and "standard deviations."

Thank you for the comment. To help the understanding of the statistical analysis we wrote out the word for ANOVA when it was first introduced in the manuscript. However, the ANOVA is a standard method in exercise science and should be familiar to the reader. Therefore, we act on the assumption that nearly all of the readers know what an ANOVA is. To explain the details of the statistical method would exceed the manuscript and would deviate from the topic of the study. (see page 6-7, lines 173-186)

3) The data are previous to the Covid-19 pandemic. A meaningful change that occurred was the adoption of 5 substitutions per match, and there is the possibility that it becomes permanent, mainly due to the health concerns of the players. And this is a good point to discuss, mainly because this work could be compared (in the future) with this new scenario of 5 substitutions.

Thank you for the comment. We added a paragraph to the limitation to address this problem and to indicate the problem for future studies. (see page 14, lines 377-380)

Reviewer #2: Report of "Center Backs Work Hardest When Playing in a Back Three: The Influence of

3 Tactical Formation on Physical and Technical Match Performance in Professional

4 Soccer" by L. Forcher et al.

The manuscript above presents a study of whether tactical formation affects the physical and 42 technical match performance of professional soccer players in the first German Bundesliga. While the topic of the manuscript is interesting, there are certain aspects that need to be clarified further, specially the methodology and robustness of the findings, before this manuscript is suitable for publication. Please find my detailed points below.

1) My main concern is that the most important results were extracted using ANOVA software, which

provides limited information, specially to evaluate the robustness of the findings. It seems that the authors

used the ANOVA as a black-box without examining details in the calculations.

Thank you for your comment and for sharing your concerns about the statistics. The ANOVA can be considered a robust method to detect arithmetic mean differences between groups. Further, the ANOVA is one of the most common methods to find significant group differences. Based on the studies mentioned underneath, the ANOVA is very robust against a violation of the pre-conditions (For example Blanca MJ, Alarcón R, Arnau J, Bono R, Bendayan R. Non-normal data: Is ANOVA still a valid option? Psicothema. 2017 Nov;29(4):552-557. doi: 10.7334/psicothema2016.383 . PMID: 29048317.). We suggest that the ANOVA as statistical method, is well studied and, therefore, should considered more likely a white-box than a black-box. In consideration of the huge number of groups and, therefore, the ANOVA represents the most robust choice in terms of statistical methods in our view. Moreover, to support the findings of the ANOVA we calculated effect sizes. The effect sizes can help the reader to interpret the differences that occurred in the ANOVA and help to interpret the robustness of the found results. In the limitations section, we discussed the problems and disatvantages with the ANOVA in the context of the present study and, therefore, ensure that the reader is capable of interpreting the results in the right way. We would be pleased, if you would share a deeper and more detailed version of your concerns. In order to further help the help the understanding of the reader we added a paragraph to the limitation section. (see page 15, lines 388-391)

2) The manuscripts present several issues regarding the organization and final presentation. How do you cite? Did you put number or names in parentheses? Lines 70, 72 and 73 (#). Names, lines 155 and 156 (names). Capital letters: lines 191, 192 and 194, figure captions; figure, table,

Thank you for making us aware of these issues. We corrected the citing and the formal aspects in the manuscript. 

3) In the introduction they do not say how the work is organized.

In our field of study (sports science), it is not common to describe how the work is organized, as original investigations commonly follow a very similar pattern. The introduction sections should introduce the reader to the topic and deduce the main research question of the study. The scientific approach to address the research question is formally attributed to the methods section.

4) The authors must consider additional metrics to present their results. The information provided by Deltatre can be used to explore additional properties, for instance, cross-correlations, etc.

Cross-correlations concern the correlation of two signals, waves, or functions. After consultation with the research group, we are unsure how to implement this kind of statistical method into the study context. We would appreciate it if you could explain in detail how you would use cross-correlations in order to evaluate the present results. Furthermore, we would be pleased if you would name and explain other statistical methods that would help to provide other statistical results that can help to fix your concerns about the present statistical analysis. 

In favor of the research question, we wanted to find the differences between different tactical formations. The correlations of the different tactical formations would analyze to what extent the performance of the players is equal/similar in the different formations. 

Because we analyzed eight different tactical formations the results section is already very large. If we would use another statistical method to describe the effect of the tactical formation, we would expand this section. To the best of our knowledge, this would not provide any further information regarding the research question and would possibly downgrade the readability of the manuscript. 

5) The authors should also consider to incorporate methodologies like complex networks to reinforce their analysis and not just use conventional statistics.

We suggested using a multi-level model to statistically evaluate our data. The mixed-model addresses inter- and intraindividual differences in the performance metrics. In favor of the research question, we wanted to exert the effect of the tactical formation and not of the individual player. Therefore, these statistical models would fit the dataset but not the research question. Therefore, we accepted and addressed the disadvantages of the ANOVA in order to properly answer the research question. We would appreciate it if you would provide more detailed suggestions in order to help us fix your concerns.

6) As a general remark, it seems that they only make a statistical analysis but not checked or confirmed with anything else. The explanation of methods is not clear at all and it is not well understood the procedure. Also the quality of figures is very poor.

We provided the figures in better quality to further help the readability of the manuscript. We tried to overhaul the statistics section in order to help the readability of the method section. . (see page 15, lines 388-391). We already addressed the pros and cons of the ANOVA. We would like to further explain to the Reviewer why we chose this statistical method. If we would consider the dependency of the groups (different tactical formations) and would suggest that the normal distribution is not given we would have to choose the Friedman-test to detect group-mean differences. Because we have eight execute different formations in ten different parameters for five different positions we have 1400 single tests. The ANOVA in SPSS controls for the alpha error with an alpha error correction (Bonferroni correction). We used Bonferroni post-hoc tests because they provide the most conservative and robust results in terms of alpha error correction. If we would execute every single group comparison independently the alpha error probability would increase significantly. Therefore, we stated the robust ANOVA with the disadvantages mentioned in the limitations as a better statistical method than dealing with a significantly increasing alpha error probability.

---

## [Decision Letter · Decision Letter 1]

3 Mar 2022

Center Backs Work Hardest When Playing in a Back Three: The Influence of Tactical Formation on Physical and Technical Match Performance in Professional Soccer

PONE-D-21-27534R1

Dear Dr. Forcher,

We’re pleased to inform you that your manuscript has been judged scientifically suitable for publication and will be formally accepted for publication once it meets all outstanding technical requirements.

Kind regards,

Haroldo V. Ribeiro

Academic Editor

PLOS ONE

Additional Editor Comments (optional):

I thank the authors for the detailed reply and for addressing all remarks of our reviewers. I considered that most issues were appropriately addressed, and a few other comments would indeed exceed the manuscript and deviate from the main topic of this work.

Reviewers' comments:

Reviewer's Responses to Questions

**Comments to the Author**

1. If the authors have adequately addressed your comments raised in a previous round of review and you feel that this manuscript is now acceptable for publication, you may indicate that here to bypass the “Comments to the Author” section, enter your conflict of interest statement in the “Confidential to Editor” section, and submit your "Accept" recommendation.

Reviewer #1: All comments have been addressed

2. Is the manuscript technically sound, and do the data support the conclusions?

Reviewer #1: Yes

3. Has the statistical analysis been performed appropriately and rigorously? 

Reviewer #1: Yes

4. Have the authors made all data underlying the findings in their manuscript fully available?

Reviewer #1: No

5. Is the manuscript presented in an intelligible fashion and written in standard English?

Reviewer #1: Yes

6. Review Comments to the Author

Reviewer #1: (No Response)

7. PLOS authors have the option to publish the peer review history of their article (what does this mean?). If published, this will include your full peer review and any attached files.

Reviewer #1: No

---

## [Editor Report · Acceptance letter]

8 Mar 2022

PONE-D-21-27534R1 

Center Backs Work Hardest When Playing in a Back Three: The Influence of Tactical Formation on Physical and Technical Match Performance in Professional Soccer 

Dear Dr. Forcher:

I'm pleased to inform you that your manuscript has been deemed suitable for publication in PLOS ONE. Congratulations! Your manuscript is now with our production department. 

Kind regards, 

on behalf of

Dr. Haroldo V. Ribeiro 

Academic Editor

PLOS ONE